# Detection of West Nile Virus Lineage 2 in Eastern Romania and First Identification of Sindbis Virus RNA in Mosquitoes Analyzed using High-Throughput Microfluidic Real-Time PCR

**DOI:** 10.3390/v15010186

**Published:** 2023-01-09

**Authors:** Luciana Alexandra CRIVEI, Sara MOUTAILLER, Gaëlle GONZALEZ, Steeve LOWENSKI, Ioana Cristina CRIVEI, Daniela POREA, Dragoș Constantin ANITA, Ioana Alexandra RATOI, Stéphan ZIENTARA, Luanda Elena OSLOBANU, Alexandru TOMAZATOS, Gheorghe SAVUTA, Sylvie LECOLLINET

**Affiliations:** 1Regional Center of Advanced Research for Emerging Diseases, Zoonoses and Food Safety, Iași University of Life Sciences, 700490 Iași, Romania; 2ANSES, INRAE, Ecole Nationale Vétérinaire d’Alfort, UMR BIPAR, Laboratoire de Santé Animale, 94700 Maisons-Alfort, France; 3ANSES, INRAE, Ecole Nationale Vétérinaire d’Alfort, UMR VIROLOGIE, Laboratoire de Santé Animale, 94700 Maisons-Alfort, France; 4Department of Arbovirology, Bernhard Nocht Institute for Tropical Medicine, 20359 Hamburg, Germany

**Keywords:** arbovirus, West Nile virus, Sindbis virus, mosquito-borne, Romania, flavivirus, alphavirus, high-throughput microfluidic real-time PCR

## Abstract

The impact of mosquito-borne diseases on human and veterinary health is being exacerbated by rapid environmental changes caused mainly by changing climatic patterns and globalization. To gain insight into mosquito-borne virus circulation from two counties in eastern and southeastern Romania, we have used a combination of sampling methods in natural, urban and peri-urban sites. The presence of 37 mosquito-borne viruses in 16,827 pooled mosquitoes was analyzed using a high-throughput microfluidic real-time PCR assay. West Nile virus (WNV) was detected in 10/365 pools of *Culex pipiens* (*n* = 8), *Culex modestus* (*n* = 1) and *Aedes vexans* (*n* = 1) from both studied counties. We also report the first molecular detection of Sindbis virus (SINV) RNA in the country in one pool of *Culex modestus*. WNV infection was confirmed by real-time RT-PCR (10/10) and virus isolation on Vero or C6/36 cells (four samples). For the SINV-positive pool, no cytopathic effectwas observed after infection of Vero or C6/36 cells, but no amplification was obtained in conventional SINV RT-PCR. Phylogenetic analysis of WNV partial NS5 sequences revealed that WNV lineage 2 of theCentral-Southeast European clade, has a wider circulation in Romania than previously known.

## 1. Introduction

In recent decades, the impact of emerging and re-emerging infectious diseases on human and animal health has increased, posing a major challenge for global health and economy [1]. Roughly two thirds of human infectious diseases originate from wildlife, and many recently emerging diseases are caused by vector-borne viruses [2]. At least 10 mosquito-borne viruses (MBV) have been detected in Europe, and some of them have become a major concern in the last decade. The majority of these MBVs belong to the *Flaviviridae* family (e.g., West Nile, Dengue or Usutu viruses) [3,4]. Other pathogenic MBVs of the families *Togaviridae* (e.g., Sindbis, Chikungunya viruses) and *Peribunyaviridae* (Tahyna, Inkoo and Lednice viruses) are known to have medical importance in Europe [5]. West Nile virus (WNV, genus *Flavivirus*) is the most widespread encephalitic arthropod-borne virus [6] and a member of the Japanese encephalitis virus serocomplex. Of its nine potential lineages documented to date [7,8], the main pathogenic WNV strains belong to lineages 1 and 2. However, human cases of acute encephalitis have also been attributed to WNV lineage 5 (formerly lineage 1c [9]) in India [10]. In the natural enzootic cycle, WNV is transmitted mainly by ornithophilic *Culex* mosquitoes as vectors and birds as amplifying reservoir hosts [9]. West Nile virus is one of the most important viral pathogens in Europe, and since its initial detection in Albania in 1958, it has become an endemo-epidemic virus on the continent [11]. Although the great majority of animal species are not susceptible to WNV infection, humans and equids are incidental hosts which may become ill without developing viraemia levels high enough for reinfecting a new mosquito during feeding. Most infections remain asymptomatic, while 1–4% lead to neuroinvasion and disease of the central nervous system [12]. Mortality is usually higher in immunocompromised or chronically ill hosts [13] and impact on public health is compounded by risks associated with organ donation and blood transfusion [14].

The currently most affected European regions include south, southeast and central Europe [15]. In recent years, consecutive vector seasons with high temperatures enabled WNV to expand its range westwards and at higher latitudes [13,14]. A sharp increase in human and equid cases has been experienced most acutely by southern and southeastern regions of the continent as a result of the rapid emergence and spread of West Nile lineage 2 virus first reported in Hungary, in 2004 [8,15]. In Romania, the 1996 epidemic caused by a lineage 1 WNV strain in southeastern Romania remains the largest one documented in the country (393 hospitalized cases and 17 deaths) [16] Since then, WNV infections of humans and equids have been recorded annually. Large outbreaks occurred also in 2010, 2016 and 2018 (277 cases), when WNV activity and impact on public health were particularly high throughout the country [17,18,19]. Circulation of WNV in Romania is endemic in southern (Romanian Plain) and southeastern parts (Constanța, Tulcea, Danube Delta) [20]. Human cases have also been detected in the last decade in the western parts of the country (Sibiu). 

Unlike WNV, the presence of Sindbis virus (SINV, genus *Alphavirus*) has been evidenced worldwide, though human outbreaks are reported almost exclusively in northern Europe (Sweden, Finland, Russia) and rarely in South Africa, China and Oceania [21,22]. This alphavirus is the etiological agent of Sindbis fever, characterized mainly by rash, arthralgia and fever. Although the illness is typically self-limiting, arthralgia and myalgia can manifest for months or even years, in rare cases resulting in chronic arthritis. Widely detected in *Culex* and *Culiseta* mosquito vectors worldwide, SINV is also [23] found in vertebrates from Eurasia, Africa and Oceania. In Europe, SINV or anti-SINV antibodies have been found in more than 20 European countries from all the main regions of the continent. The presence of SINV in Romania remains doubtful, as animals were reportedly found seropositive in the mid-1970s by Drăgănescu et al. [24] at very low frequency. 

Despite endemic circulation of WNV and high incidence of WNV infections in humans, Romania still lacks an operational integrated surveillance program for arboviruses. This is also reflected in the scarcity of genetic and serological data, hampering molecular and genomic epidemiological studies in the region. Obtaining baseline data about arbovirus spatial distribution, diversity, prevalence in vectors and impact on hosts are necessary steps for pursuing regional and country-wide integrated surveillance. Therefore, the aims of this study were to investigate the presence and diversity of MBVs in two counties from eastern and southeastern Romania: one which is known for endemic circulation of WNV (Tulcea) and one where sporadic WNV cases occurred in recent years (Iași). 

## 2. Materials and Methods

### 2.1. Study Areas and Mosquito Sampling

Mosquitoes were collected during two consecutive transmission seasons (May–September, 2018 and 2019) in two counties from eastern (Iași) and southeastern (Tulcea) Romania. Collection sites were selected by the apparent suitability of habitats for MBV transmission, including the abundance of vertebrate hosts. Mosquitoes were collected using a combination of CDC Light Trap Miniature traps (BioQuip, Rancho Dominguez, CA, USA), BG-Sentinel traps (BioGents, Regensburg, Germany), manual aspirator “Pooter” Style (BioQuip, Rancho Dominguez, CA, USA) and the manual aspirator Heavy Duty Hand-Held DC Vac (BioQuip, Rancho Dominguez, CA, USA).

Sampling in Iași was conducted in three urban sites, one peri-urban site (a private household) and one natural site (game reserve). The urban sampling sites were highly anthropogenic, characterized by a mix of wooded areas (campus, parks), water bodies, high density of bird hosts and availability of artificial resting and breeding sites (car tire depots and old buildings). The peri-urban and natural sites were chosen based on ecological characteristics favorable to MBV transmission: wetland, density of wildlife hosts including birds, abundance of vectors (game reserve), domestic mammal hosts and synanthropic birds (private household) (Figure 1).

Two sites were selected in Tulcea county for opportunistic collection of mosquitoes in July and September 2019. The first site was located in an urban area comprising residential/commercial complexes. The second site was located in a natural area with a relatively low degree of anthropization, adjacent wetlands and a horse shelter.

### 2.2. Mosquito Identification and Processing

Mosquitoes were brought alive to the laboratory in catch nets and killed by deep-freezing for 5 min, at −20 °C. Females were sorted according to gonotrophic stage (unfed, fed, gravid) and identified on chilled tablets using morphological keys [25]. Unfed mosquitoes were pooled according to the taxa, sex, gonotrophic stage, collection date and site. After identification, monospecific and monogeneric pools (of individuals which could not be identified below the genus level), were transferred to −80 °C until viral screening. Blood-fed females were screened individually, while the unfed mosquito pools were analyzed in monospecific pool, each containing 20–50 specimens (“group testing”). 

### 2.3. RNA Extraction 

Mosquito pools were homogenized at 5500 rpm in 2 mL tubes containing silica beads (0.1 mm diameter, BioSpec, Bartlesville, OK, USA) and 500 μL of DMEM with 10% fetal calf serum on a Precellys 24 Dual homogenizer (Bertin, Montigny-le-Bretonneux, France). The homogenate was clarified by centrifugation for 2 min at 1.500 rpm and 120 μL of the supernatant was used for RNA extractions using the MagVet Universal Isolation kit (Lifetechnology, MA, USA) and MagMAX Express-96 Deep Well Magnetic Particle Processor workstation (Thermo Fisher Scientific, MA, USA). The remaining homogenate was stored at −80 °C for subsequent virus isolation. 

### 2.4. Reverse Transcription and cDNA Pre—Amplification

Reverse transcription of RNA extracts was performed using Reverse Transcriptase Master Mix (Starlab Biotools, Hamburg, Germany). One μL extracted RNA was added with 1 μL of Reverse Transcription Master Mix to 3 μL of RNase free water, resulting in a total reaction volume of 5 μL. The reaction’s thermal profile consisted of one cycle of 5 min, at 25 °C, one step of 30 min, at 42 °C, and one last cycle, at 85 °C, for 5 min.

Pre-amplification reactions were performed using the PreAmp Master Mix Kit (Starlab Biotools, Hamburg, Germany), following the manufacturer’s instructions. Pre-amplification allowed a better amplification of viral cDNA relative to mosquito cDNA. The pre-amplification reaction used 1 μL of the master mix, 1.25 μL of cDNA, 1.25 μL of pooled primers (all the primers targeting arboviruses) and 1.5 μL distilled water. This operation was conducted using the following thermal conditions: 95 °C for 2 min, 14 cycles at 95 °C for 15 s, and 4 min at 60 °C. After the pre-amplification step, cDNAs/amplicons were diluted 1:5 in pure water and stored at −20 °C until further analysis.

### 2.5. High-Throughput Microfluidic Real-Time PCR

Mosquito-borne viruses, their targeted genes and the corresponding primers/probe sets were selected according to Moutailler et al. [26]. For this study, 365 field-collected mosquito pools were screened for 37 viruses, targeting a total of 94 genes (Appendix A). 

The pre-amplified cDNA was subjected to high-throughput microfluidic real-time PCR amplification using the 96.96 dynamic arrays (Starlab Biotools, Hamburg, Germany). For the purpose of this assay, the BioMark real-time PCR system (Starlab Biotools, Hamburg, Germany) was used. High-throughput real-time PCRs were performed using FAM and black hole quencher (BHQ1)-labeled TaqMan probes with TaqMan Gene Expression Master Mix in accordance with manufacturer’s instructions (Applied Biosystems, MA, USA). The reactions were performed for 2 min at 50 °C and 10 min at 95 °C, followed by 40 cycles of two-step amplifications of 15 s at 95 °C and, finally, 10 min at 60 °C. 

Data were acquired on the BioMark real-time PCR system and analyzed using the Fluidigm real-time PCR Analysis software to obtain cycle threshold values, which for positive samples ranged between 16.3 and 26.5. False positive or inconclusive (unreliable) results were declared at Ct > 30. Primers and probes were evaluated before use for their specificity against cDNA reference samples (see [26] for details). One negative water control was included per chip. To determine if factors present in the sample could inhibit the PCR, *Escherichia coli* strain EDL933 DNA was added to each sample as an internal inhibition control, using primers and probes specific for the *E. coli eae* gene [26].

### 2.6. Validation by Virus Isolation and Partial Genome Sequencing

WNV-positive pools detected by microfluidic real-time PCR assays were further screened by WNV real-time PCR using Applied Biosystems StepOnePlus Instrument as described by Linke et al. [27]. The quantitative duplex real-time PCR used FAM–TAMRA-labeled probes for WNV and VIC–TAMRA for the endogenous β-actin RNA, as described in [28] (Appendix A). An initial reverse transcription step was performed at 45 °C for 10 min, followed by DNA denaturation at 95 °C for 10 min, and 45 PCR cycles at 95 °C for 15 s and 60 °C for 1 min.

After confirmation of WNV positive samples by duplex real-time PCR, virus isolation was performed on freshly prepared semi-confluent monolayers of Vero cells (ATCC CCL81) maintained in Dulbecco′s Modified Eagle′s Medium (DMEM) supplemented with 1% L-glutamine, 1% non-essential amino acids, 1% penicillin/streptomycin, 1 mM sodium pyruvate and 10% fetal bovine serum (FBS), at 37 °C and 5% CO_2_. T25 flasks of Vero cells were inoculated with 1 mL inoculum, consisting of 100 μL of 0.45 µm-filtered mosquito homogenate and 900 μL DMEM. Seven days post-infection (dpi), cell supernatants were recovered, aliquoted and stored at −80 °C for further passage in cell culture, while RNA was extracted using QIAmp Viral RNA Mini QIAcube Kit (Qiagen, Hilden, Germany) and the QIAcube robot (Qiagen, Hilden, Germany), following the manufacturer’s instructions.

Virus supernatants recovered from infected Vero cells were further propagated on C6/36 cells (ATCC CRL-1660). An amount of 1 ml of supernatants was added on confluent monolayers of C6/36 cells. *Aedes albopictus*-derived C6/36 cells were maintained at 28°C in Leibovitz’s L-15 medium, supplemented with 10% FBS, 1% L-glutamine, 1% non-essential amino acids, 1% penicillin/streptomycin and 1 mM sodium pyruvate. The supernatant was harvested at 7 dpi, and C6/36 RNA extracts were checked for the presence of WNV RNA by real-time PCR.

The positive SINV pool detected by microfluidic real-time PCR assays was further screened by conventional SINV PCR, as described by Lundström and Pfeffer [29], and viral isolation was attempted on Vero and C6/36 cell cultures (Appendix A).

### 2.7. Calculation of Minimum Infection Rates

Considering that the expected prevalences of WNV and other MBVs in mosquitoes are lower than 0.1% in Europe and assuming that generally a single individual was infected within a positive pool, prevalences were expressed as the minimum infection rate per 1000 tested individuals (MIR), by calculating the ratio of the positive pools to the total mosquitoes tested. 

### 2.8. Sequencing and Phylogenetic Analysis of WNV

Amplification of partial NS5 gene by conventional pan-flavivirus PCR was performed on positive mosquito pool homogenates and virus isolates, as described by Weissenböck et al. [30]. 

Amplification was achieved under the following thermal cycling conditions: 30 min at 50 °C, 15 min at 95 °C, 45 cycles at 94 °C for 30 s, 60 °C for 30 s, 72 °C for 1 min and 10 min at 72 °C. Resulting amplicons were visualized by electrophoresis (30 min at 90V) on a 1% agarose gel in the presence of a 1kb ladder (Hyperladder, Bioline, London, UK).

Sanger sequencing was performed in both directions and the resulting nucleotide sequence was identified by basic alignment search tool (BLAST) in the Genbank database (https://blast.ncbi.nlm.nih.gov/, accessed on 30 November 2022). Homologous sequences were retrieved and aligned using the MAFFT algorithm implemented in Geneious Prime (Biomatters, Auckland, New Zealand). To assess the phylogenetic relationship of the new WNV isolates, a maximum likelihood phylogenetic tree rooted by WNV lineage 3 (Rabensburg strain) was built with MEGA 11 [31] following substitution model selection by Mega 11 and jModelTest 2.1.10 [32]. Robustness of tree nodes was assessed by 1000 bootstraps, and the result was displayed with iTOL v5 [33].

## 3. Results

### 3.1. Mosquito Trapping 

Between June 2018 and September 2019, we collected 17,694 female mosquitoes representing 11 taxa in four urban, one peri-urban and two natural sites from two counties in eastern and southeastern Romania. From this collection, seven taxa which are known to be MBV vectors (including the *Aedes* spp. pools) were screened using a high-throughput microfluidic real-time PCR targeting 37 viruses from 4 families (*Peribunyaviridae*, *Flaviviridae*, *Reoviridae*, *Togaviridae*).

*Aedes vexans* (43.77%) and *Culex pipiens* s.l. (29.21%) were the most commonly detected of the 11 mosquito taxa identified (Figure 2). The rest of the identified taxa represented between 0.03 % and 7.34% of the mosquito collection.

The largest number of individuals was captured in the natural sites from the two counties, with 8727 specimens (49.3%) collected in Iași and 6863 (38.8%) in Tulcea. Consequently, in these sites, we recorded the highest taxonomic diversity, amounting to nine taxa in the natural site from Iași (natural site 1) and eight taxa in the natural site from Tulcea (natural site 2). The largest sample in urban environments was obtained in Iași, in urban site 1 (1286 specimens from three taxa), followed by urban site 2 (551 specimens from three taxa), urban site 3 (10 specimens of *Culex pipiens* s.l.) and urban site 4 in Tulcea (4 individuals from three taxa) (Figure 2). The majority of the identified mosquitoes were collected in Iași county (*n* = 10,827, 61.2%, versus 6,867 individuals from Tulcea, 38.8%). *Aedes vexans* and *Culex pipiens* s.l. were the dominant taxa in the majority of sampling sites and the overall collection (Figure 2). Exceptions were found in the urban sites 2 and 3 from Iași and urban site 4 from Tulcea, where *Culex pipiens* s.l. was the main taxon and *Aedes vexans* were absent. 

From the total collection, 6 taxa of the *Aedes* and *Culex* genera, along with the unidentified *Aedes* spp. individuals, (*n* = 16,827, 95.1%) were pooled (*n* = 365) and subjected to virus screening by high-throughput microfluidic real-time PCR.

### 3.2. Virus Detection

#### 3.2.1. Detection of Viral Pathogens Using High-Throughput Microfluidic Real-Time PCRs

Through the screening of 37 MBVs using high-throughput microfluidic real-time PCRs, a total of 11 out of 365 (3%) mosquito pools were found positive for WNV (10 pools 2.73%) and SINV (1 pool, 0.27%) (Figure 3). 

Three mosquito taxa were found to be WNV positive: *Culex pipiens* s.l. (eight pools), *Culex modestus* (one pool) and *Aedes vexans* (one pool) (Table 1). The majority of WNV-positive pools originated from natural site 1 (Iași, 5/127), followed by natural site 2 (Tulcea, 3/127) and urban site 2 (Iași, 2/12). *Culex pipiens* s.l. was found to be WNV positive in each month from June to September (2018) in both natural sites (1 and 2) and urban site 2. In all these sites, the habitat consisted of wooded landscape and wetland. *Culex modestus* positive for WNV was detected in August 2019 in urban site 2. Positive *Aedes vexans* were collected in July 2019 at the natural site 2, from Tulcea. Sindbis virus was detected in a single pool of *Culex modestus* sampled in June 2018, at the natural site 1, in Iași (Figure 3).

Overall, the number of WNV-positive pools was similar in 2018 (*n* = 4) and 2019 (*n* = 6), respectively. The minimum infection rate (MIR) for WNV ranged between 0.34 for *Aedes vexans* to 3.54 for *Culex pipiens* s.l. in Iași in 2018. The MIR for SINV-positive *Culex modestus* was 1.33 (Table 2).

#### 3.2.2. Validation of Pathogen Detection by Monospecific Real-Time RT-PCR and Virus Isolation

Confirmation of WNV infection was carried out on positive pools evidenced in high-throughput microfluidic assays by real-time RT-PCR and by virus isolation. The real-time RT-PCR assay targeted a different fragment of the WNV genome (e.g., the 5′ non coding region and part of the capsid coding sequence) [27]; Ct values ranging between 20.7 and 32.6 were obtained for every WNV-positive pool (Appendix A).

Successful virus isolation, as assessed by observation of cytopathic effects (CPE), was obtained in 4 of 10 inoculated Vero cell cultures and for one sample in C6/36 cells. 

The *Culex modestus* pool found positive by real-time microfluidic SINV PCRs did not induce CPE on Vero and C6/36 cells, nor was positive in conventional SINV RT-PCR [29]. 

### 3.3. Phylogenetic Analysis 

Two partial NS5 sequences were obtained from positive pools of *Culex pipiens* s.l. Bioinformatic analysis of these sequences indicated that both mosquito homogenates, originating from *Culex pipiens* s.l. pools collected on the 14th of September 2018 and 11th of August 2019, respectively, were infected by WNV lineage 2 strains of the Central-Southeast European clade (Figure 4).

## 4. Discussion

Vector-borne diseases are a growing concern worldwide, especially against a backdrop of rapid environmental changes, such as rapid urbanization, expanding networks for the mobility of goods and people, as well as changing climatic patterns. These factors favor expansion of MBVs toward higher latitudes in Europe, as conditions become more suitable for the mosquito vectors to colonize new regions. Furthermore, growing temperatures support virus transmission by supporting vector population reproduction and shortening extrinsic incubation periods. This is especially the case of WNV, which, in the last two decades, has established endemo-epidemic transmission in new regions of Europe [15,34]. Concomitantly, recurring autochthonous outbreaks of DENV in southern France, northern Italy and Croatia [35,36,37] highlighted the high introduction risk associated with MBVs transmitted by *Aedes* mosquitoes. 

Most mosquito taxa were found by our study in the natural sampling sites from Iași and Tulcea, where ecological characteristics seem to support higher vector diversity. *Aedes vexans* were found positive for WNV for the first time in Romania. Studies of vector competence have shown the ability of this species to transmit WNV [38,39]. However, its mammophilic character suggests that *Aedes vexans* could play only a minor role in WNV transmission, unlike the ornitophilic/generalist *Culex pipiens* s.l., which is the main WNV vector and was repeatedly found infected in natural and anthropic areas from the east and southeast of the country [40,41,42]. Previous investigations of WNV epidemic activity have shown that the bioforms of *Culex pipiens* are thriving in urban settings where reservoir host populations reach high densities and particular urbanization features enable high mosquito densities near human habitation [43,44]. Hence, the detection of WNV in *Culex modestus* and *Culex pipiens* s.l. from the urban site 2 from Iași warrants the implementation of surveillance in urban and peri-urban settings. 

The prominent role of *Culex pipiens* s.l. as a WNV vector in the studied areas is also indicated by the overall MIR. The same can be observed for *Aedes vexans*, whose secondary role is reflected by the lowest MIR found in the present study. The higher prevalence of WNV observed in our study in 2018 was also reflected by the high number of human cases reported in Iași, suggesting that high levels of enzootic WNV circulation resulted in more frequent spill-over to humans during this vector season. This epidemiological scenario was observed in 2018 by other European countries where incidence rates in humans grew sharply along with new reports of infections with the closely related Usutu virus [45,46,47,48]. In parallel with increasing incidence in endemic regions, during the unusually hot seasons of 2018 and following years, WNV expanded its range into northwestern Europe [49,50], where epizootics affecting birds and equids were followed the next season by WNV epidemics [49].

Following the large epidemic of 1996 in southeastern Romania, the causative WNV lineage 1 strain was replaced by WNV lineage 2 strains of the Eastern European clade (EEC, Danube Delta, Volga Delta, Ukraine) and the Central-Southeastern European clade (CSEC)**.** Lineage 2 was first detected in Romania in 2010, as closely related to the 2007 Volgograd outbreak strain (EEC).

The epidemiological data showed that the EEC strains dominate transmission in Romania from 2010-2015 in mosquitoes collected in the Danube Delta [41] and Bucharest while CSEC strains were detected from 2015 onwards and had replaced earlier EEC lineage 2 strains by 2016 [51]. Lineage 2 strains had not been evidenced previously in Iași county, although WNV infections in humans are reported annually, and sustained WNV circulation has been evidenced through prevalence studies in horses [20,42] and dogs [42,52]. 

Bird migration is considered one of the main mechanisms of WNV introduction and spread [34]. For Europe, intercontinental flight along the main corridors of the Afro-Eurasian migration network is generally accepted as a mechanism of introduction for new WNV strains, and it may play a significant role at major migratory hubs. However, new introductions from Africa are infrequent, and WNV relies more on short-range movements of infected birds where local mosquito populations acquire the virus for local transmission. The presence of the virus is also associated with intensive farming, mammal richness, urbanization and wetland coverage [53]. These environmental factors are conducive to viral transmission as they support rich and dense communities of hosts and mosquito vectors. Our results reflect this scenario, as we found WNV-positive mosquito pools in both natural and urban sites sharing key ecological features (water bodies, high density of birds and wooded areas).

Of the nine endemic MBVs known in Europe, WNV, USUV and SINV are transmitted by *Culex* mosquitoes [54]. The widespread presence of SINV in Europe was assessed mainly by serological methods [21], and for some countries, the data are outdated. Isolation of SINV has been performed to date only from mosquitoes collected in Sweden, Norway, Finland, Germany, and Russia [55]. Recent serosurveys conducted in Romania on migratory birds [56] did not confirm the SINV activity reported by Draganescu et al., in the 1970s [24]. Despite the SINV RNA detection, we could not isolate SINV, nor amplify it by PCR, thus making it unclear whether we detected only SINV genome fragments. This issue represents the main limitation of the study, and further studies are needed to assess the status of SINV in Romania. 

The incidence of West Nile virus infection has increased in Romania and other European countries in the last decade, as well as the geographic range of the virus. The integrated surveillance approach used by several European countries is much needed in a country such as Romania where WNV is firmly established. Annual counts of human and equid infection in Romania in 2010–2018 were among the highest in Europe, surpassed only by those of Italy, Serbia and Greece, countries which have integrated surveillance programs [57]. The current study provides insights into the viral diversity and circulation of MBVs in eastern/southeastern Romania. The CSEC clade of WNV was found to have a wider distribution in Romania than previously known. Despite the first molecular detection of SINV in the country, our results do not conclusively show its establishment in Romania. Such preliminary data, however, seem to indicate that a wider array of MBVs than the ones historically reported could be present in Romania and should promote continuous sampling efforts, surveillance and molecular characterization of MBVs in the country.

## Figures and Tables

**Figure 1 viruses-15-00186-f001:**
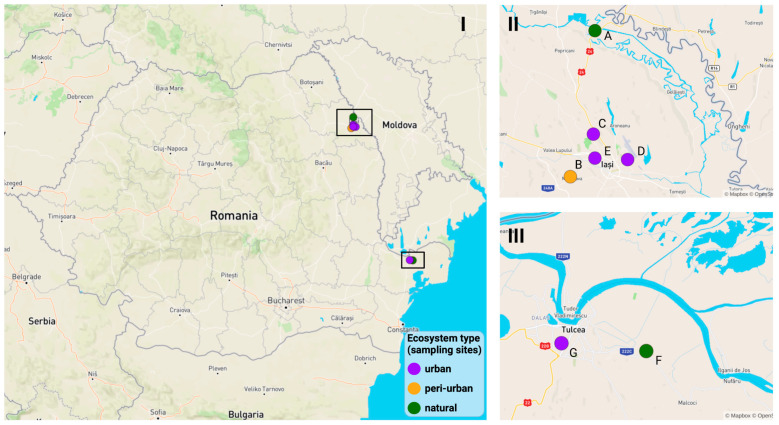
Mosquito sampling sites in eastern Romania used in this study (2018–2019) (**I**). Upper-right panel shows the sampling sites in Iași county (**II**) and the lower-left panel indicates the trapping locations in Tulcea county (**III**). Purple circles represent urban sites, green circles represent natural sites and the orange circle indicates peri-urban sampling sites. Natural site 1 = A, peri-urban site 1 = B, urban site 1 = C, urban site 2 = D, urban site 3 = E, natural site 2 = F, urban site 4 = G.

**Figure 2 viruses-15-00186-f002:**
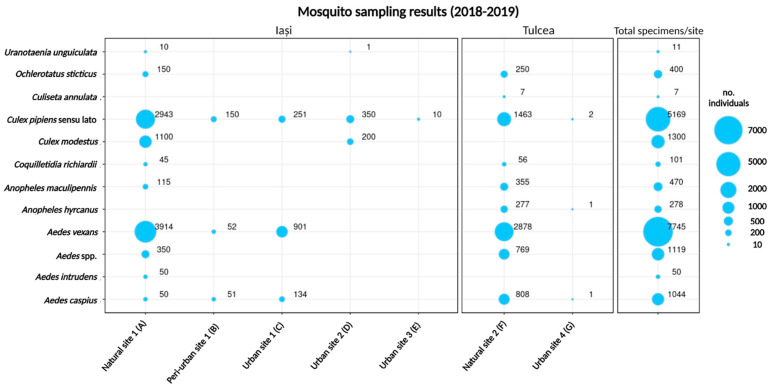
Number of mosquito specimens trapped in 2018 and 2019 in eastern Romania, ordered by taxa and sampling site (Iași county in the left panel, Tulcea in the central panel, and total per taxa on the right panel).

**Figure 3 viruses-15-00186-f003:**
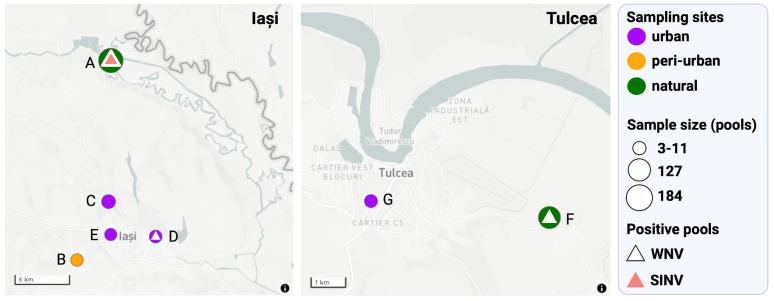
Mosquito screening results relative to the distribution of sampling sites in the two counties in eastern Romania. Purple circles represent urban sites, green circles represent the natural sites and the orange circle indicates the peri–urban sampling site. Natural site 1 = A, peri-urban site 1 = B, urban site 1 = C, urban site 2 = D, urban site 3 = E, natural site 2 = F, urban site 4 = G.

**Figure 4 viruses-15-00186-f004:**
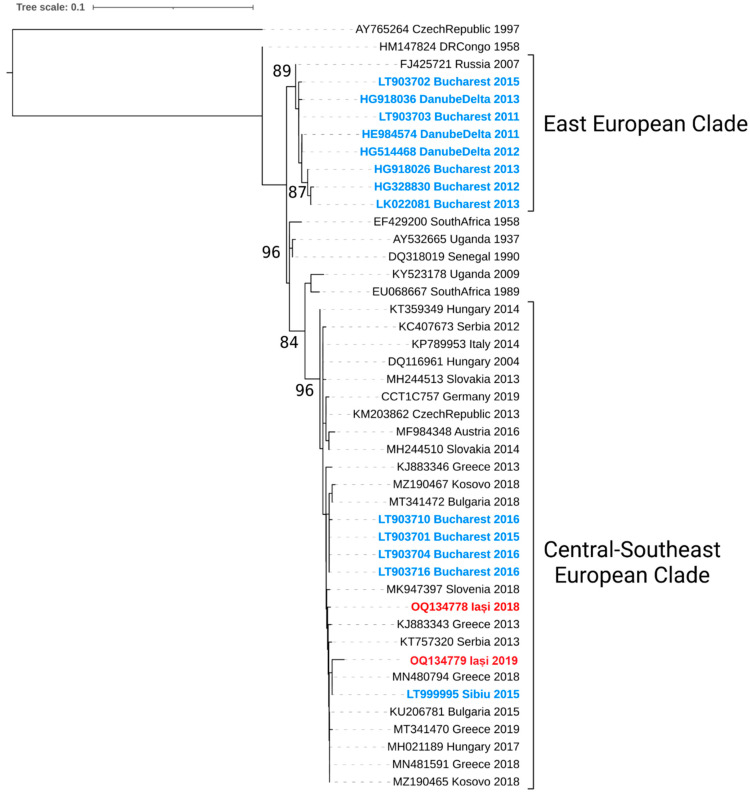
Maximum likelihood phylogenetic analysis of WNV lineage 2 partial NS5 gene. Romanian strains are indicated in blue (previous studies) and red color (present study). Rabensburg strain 97–103 was used as an outgroup, and bootstrap support >80% is displayed at nodes of major clades.

**Table 1 viruses-15-00186-t001:** Virological findings in mosquitoes collected in Iași and Tulcea counties, in 2018 and 2019. The number of tested mosquitoes (Nt) and tested pools (NP) and WNV and SINV positive pools are indicated.

Species	NP	Nt	WNV+	Date of Collection of Positive Pool	Site	SINV +	Date of Collection	Site
*Aedes vexans*	167	7745	1	9 July 2019	Natural site 2 (F)			
*Culex pipiens* s.l.	116	5169	8	19 August 2018	Natural site 1 (A)			
				8 September 2018	Natural site 1 (A)			
				23 August 2018	Natural site 1 (A)			
				14 September 2018	Natural site 1 (A)			
				14 August 2019	Urban site 2 (D)			
				9 July 2019	Natural site 2 (F)			
				10 July 2019	Natural site 2 (F)			
				11 August 2019	Natural site 1 (A)			
*Culex modestus*	26	1300	1	30.08.2019	Urban site 2 (D)			
						1	14 June 2018	Natural site 1 (A)
*Aedes caspius*	24	1044	0					
*Ochlerotatus sticticus*	8	400	0					
*Aedes* spp.	23	1119	0					
*Aedes intrudens*	1	50	0					
	365	16,827	10			1		

**Table 2 viruses-15-00186-t002:** MIR by WNV for *Culex pipiens* s.l., *Culex modestus* and *Aedes vexans* mosquitoes, per time period (2018 or 2019) and county (Iași or Tulcea).

		2019			2018	
		*Culex pipiens* s.l.	*Culex modestus*	*Aedes vexans*	*Culex pipiens* s.l.	*Culex modestus*
Iași	Total WNV-positive pools	2	1	0	4	0
	Total analyzed specimens	2576	550	4460	1128	750
	MIR WNV	0.77	1.81	0.00	3.54	0.00
	Total SINV- positive pools	0	0	0	0	1
	Total analyzed specimens	2576	550	4460	1128	750
	MIR SINV	0.00	0.00	0.00	0.00	1.33
Tulcea	Total WNV-positive pools	2	0	1		
	Total analyzed specimens	1465	0	2878		
	MIR WNV	1.36	0.00	0.34		

## Data Availability

The sequences are submitted in the GenBank database under accession numbers OQ134778 and OQ134779.

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
