# Peer review of "Detection of West Nile Virus Lineage 2 in Eastern Romania and First Identification of Sindbis Virus RNA in Mosquitoes Analyzed using High-Throughput Microfluidic Real-Time PCR"

_viruses, 2023, doi:10.3390/v15010186_

Round 1

Reviewer 1 Report

Overall, the text provides a good overview of the importance of mosquito-borne viruses in Europe and the focus of the study on two particular counties in Romania. However, more detail and context would be helpful in fully understanding the significance and implications of the research.

Author Response

Please find the response to the Reviewer’s comments in the attached file.

Thank you! 

Reviewer 2 Report

The article “Detection of West Nile virus lineage 2 in eastern Romania and first identification of Sindbis virus RNA in mosquitoes analysed using high throughput microfluidic real time PCR” screened for the presence of Mosquito borne viruses in two county sites in Romania. This is an important paper as vector borne viruses are on rise due to climate change, resistance and urbanization. One of the significant findings of the paper is the incrimination of Aedex vexans for WNV. Europe is a special focal point and area of interest as WNV is continuous to emerge in the areas. Although paper is nicely presented and thoroughly written I have included few observations which may help authors.

Introduction

1. The authors mentioned somewhere in line 12 that only Lineage 1 & 2 are reported to be pathogenic for humans and animals. This is not necessarily true and I request authors to please check pathogenicity of lineage 5 WNV.

2. Please put a reference for nine potential lineages. This is important as few research group only consider 2 WNV lineages and others to be sub-clade.

3. Although mosquito collection sites were choosen in a meticulous way, still they represent very limited sites. Any reasons why surveillance was done in limited areas.

Materials and Methods

4. T25 flask generally inoculated with 5 ml media (inoculums). Please explain why 1 ml inoculums were administered.

Results

5. Any possible explanation, why virus can be isolated from only 4 inoculums out of 10 administered into cell lines. This may be explained in discussion section.

6. Although SINV RNA is detected, still it has not established its presence in Romania, continuous surveillance will be important for its detection and subsequent characterization.

Author Response

(The authors gave the same response as above.)
